# Constructs across a hierarchical, dimensional model of psychopathology show differential associations with social and general cognitive ability

Scott D. Blain[1,2]*, Jerillyn S. Kent[3], Timothy A. Allen[4], Carly A. Lasagna[5], Chloe A. Peyromaure de Bord[3], Aisha L. Udochi[6], Scott R. Sponheim[6,7‡], Colin G. DeYoung[7‡], Ivy F. Tso[1,2,5‡]

1 Department of Psychiatry and Behavioral Health, The Ohio State University, Columbus, OH, United States of America, 2 Department of Psychiatry, University of Michigan, Ann Arbor, MI, United States of America, 3 Department of Psychology, The University of Texas at Dallas, Richardson, TX, United States of America, 4 Department of Psychiatry, University of Pittsburgh, Pittsburgh, PA, United States of America, 5 Department of Psychology, University of Michigan, Ann Arbor, MI, United States of America, 6 Department of Psychology, University of Minnesota Twin Cities, Minneapolis and Saint Paul, Minnesota, United States of America, 7 Department of Psychiatry, University of Minnesota Twin Cities, Minneapolis and Saint Paul, Minnesota, United States of America

‡ SRS, CGD, IFT are contributed equally and share senior authorship of the paper.
* Scott.blain@osumc.edu

**Data Availability Statement:** Data from Sample 1 will be made publicly available on the NIMH Data Archive and through the Connectome Coordination

## Abstract

Many psychiatric disorders and associated psychopathology dimensions are related to social cognitive deficits and reduced general cognitive ability. The current study applied a hierarchical, dimensional approach to better understand associations among psychopathology, social cognition, and general cognitive ability. Data were collected from two samples (n = 653), including psychosis-spectrum patients, their first-degree relatives, and individuals from community sources. Participants completed dimensional psychopathology measures and social cognition tasks (e.g., emotion perception and mentalizing). Data were analyzed using bi-factor exploratory structural equation modeling. Detachment—a psychopathology dimension conceptually linked to social functioning—was associated with worse social cognition, independent of general cognitive ability. Eccentricity and Machiavellianism were associated with better social cognition and general cognitive ability. Findings—and the hierarchical, dimensional approach employed—will be useful in informing future research on and interventions for social dysfunction.

## Introduction

Humans are inherently social, yet there is significant variation in our ability and motivation to form and maintain relationships. Some individuals navigate complex social situations—like networking, dating, and interviewing—with ease, while others find even simple interactions—such as visiting a coffee shop or making a phone call—a challenge. Investigating individual

Facility. Data from Sample 2 cannot be made openly available to the public, due to wording in our original institutional review board protocol and consent materials. However, relevant materials, measures, and analytical code are stored on the Open Science Framework. Unrestricted materials will be made publicly available, and readers can request access to private materials (e.g., data from Sample 2) by reaching out to the authors. Our OSF repository can be found at the following link: https://osf.io/d6r2b/.

**Funding:** This research was supported by the National Institute of Health (U01MH108150 to S.R. S., R03DA029177-01A1 to C.G.D., R01MH122491-01 to I.F.T.) and National Science Foundation (SES-1061817 to C.G.D. and Graduate Research Fellowships to S.D.B., C.A.L., and A.L.U.).

**Competing interests:** The authors have declared that no competing interests exist.

differences in the mechanisms that facilitate social interaction provides insight into this variation. For instance, social cognitive processes—including emotion recognition and mentalizing—enable individuals to recognize and interpret the mental and emotional states of others. These processes are crucial for adaptive functioning; however, individuals with mental illness or at higher risk of psychopathology often experience difficulties in one or more of these social cognitive domains.

Research has consistently reported social cognitive deficits across a broad range of mental disorders—including schizophrenia, personality disorders, anxiety, and depression—and autism spectrum disorder [1–5]. Indeed, the National Institute of Mental Health recognizes social processes as one of five core domains in its Research Domain Criteria approach, an initiative that seeks to investigate and understand mental illness as a function of core cognitive and affective processes rather than discrete entities [6].

Diagnostic categories have long been the main method of organization in psychopathology research, but empirically, psychopathology is more accurately represented in terms of continuous dimensions [7–9]. Newer approaches to nosology, such as the Hierarchical Taxonomy of Psychopathology (HiTOP) model, suggest that psychopathology has a hierarchical structure with correlated dimensions ranging from broad *superspectra* (e.g., the p-factor or general factor of psychopathology) to more specific *spectra* (e.g., thought disorder, antagonism, disinhibition, detachment, or internalizing) and finally more specific traits and symptoms (e.g., callousness, anhedonia, or manipulativeness) [10–12]. Dimensions at various levels of the psychopathology hierarchy appear to be related to social cognition. To improve our understanding of how social cognitive deficits might contribute to psychopathology and functional impairment, it is important to clarify the associations of such deficits with various psychopathology dimensions. Such work could inform future research on the development of interventions targeting specific forms of psychopathology.

To our knowledge, there have been no published studies explicitly examining associations between the p-factor and social cognition. Beyond the p-factor, antagonism and psychotic psychopathology (or elevated psychosis-related traits) have each been tied to social cognitive deficits. Research on antagonistic traits (i.e., tendencies toward disagreeable or antisocial behavior) and social cognition has been one major focus of the *dark triad* literature, which addresses traits of psychopathy, narcissism, and Machiavellianism [13]. Psychopathy and related tendencies toward aggression show negative associations with social cognition, whereas associations with narcissism and Machiavellianism are either near-zero or positive [14–16]. Social cognition research is also prominent in the literature on *schizotypy*, which is typically conceptualized as a multidimensional set of subclinical traits related to schizophrenia and psychotic psychopathology [17]. Schizotypy is usually divided into positive, negative, and disorganized dimensions, which parallel the corresponding symptom clusters of schizophrenia; other distinct sub-dimensions have also been discussed, including eccentricity, impulsive nonconformity, and suspiciousness or paranoia [18, 19]. Positive, negative, and disorganized dimensions of schizotypy have all been linked to social cognitive deficits [20–22].

Despite existing in independent literatures (and using their own associated questionnaires), dimensions related to *schizotypy* and the *dark triad* can also be captured using broader measures of dimensional psychopathology, such as the Personality Inventory for DSM-5 (PID-5) [23–26]. For example, the dark triad traits can be mapped onto lower-level facets of the PID-5 *antagonism* domain, positive schizotypy overlaps with the *psychoticism* domain, and negative schizotypy overlaps with *detachment*. At least two studies have used dimensional measures like the PID-5 to examine individual differences in social cognition related to these domains [14, 27]. Finally, in addition to work related to schizotypy and antagonism, there is also evidence linking poorer social cognition to higher levels of negative affect [28].

Though various psychopathology dimensions and related diagnoses have been studied in association with social cognitive deficits specifically, general cognitive ability may also play a role in the relation between psychopathology and social cognition. There is a fairly strong positive correlation between performance on tests of general cognitive ability and measures of social cognition or similar constructs such as emotional intelligence [14, 29–31]. Moreover, many of the same psychopathology dimensions and diagnoses that have been linked to social cognitive deficits—particularly psychosis and associated dimensional constructs—have also been linked to broader cognitive deficits [32–35]. Likewise, several studies have shown the p-factor is negatively associated with general cognitive ability [10, 36, 37]. Thus, it is important for researchers hoping to elucidate relations between social cognition and psychopathology to also consider and test for the potential role of general cognitive ability. This would help to reveal what proportion of the relations between social cognition and psychopathology dimensions are due to general cognitive deficits versus social cognition specifically.

Links between psychopathology and social-cognitive functioning have been frequently studied, but results have been inconsistent. One reason for this inconsistency may be due to variability in the experimental tasks used to measure social and general cognition. Single-task performance-based indicators are often limited in scope and can measure constructs that are either broader or narrower than intended [38, 39]. For instance, performance on a theory of mind task might be partially driven by variance related to non-social processes like verbal ability, working memory, or perceptual acuity; conversely, a study of social cognition might be restricted to a single task measuring facial affect recognition. Relying on multiple measures likely will yield more valid measures of constructs and is further strengthened through using latent variable methods such as structural equation modeling (SEM) [33, 39, 40]. Latent variables represent the shared variance of multiple measured (or *manifest*) variables; assessing variables of interest at the latent level allows for more robust conclusions, as latent variables capture only the shared variance of indicators, thereby eliminating task-specific and error variance [41]. Modeling social cognition as shared variance in performance across tasks should give a better representation of true variance in social cognitive ability by factoring out unique task variance (which includes task-specific variance and measurement error). One can then examine how factors (e.g., general and social cognitive ability) are related to other constructs of interest, such as general and specific psychopathology dimensions.

Although several articles have specifically discussed the ubiquity of social cognitive deficits across psychiatric disorders, the potential role of general cognitive ability in these deficits, and how different disorders might be associated with specific deficits [42–44]—empirical work on these topics remains limited, especially when it comes to integration with hierarchical models like HiTOP. The current study integrated measures of social and general cognitive performance with hierarchical, dimensional models of psychopathology to (1) investigate the associations between social cognition and both general psychopathology and specific psychopathology dimensions and (2) empirically test the potential role of general cognitive ability in the psychopathology-social cognition association. The current study approached these questions using multiple datasets spanning the general population, psychiatric patients, and their first-degree relatives; these datasets included one sample with elevated levels of thought disorder/detachment and a sample with relatively high levels of externalizing (i.e., disinhibition and antagonism).

We hypothesized that a strong general factor would emerge from the psychopathology scales—along with specific factors roughly corresponding to dimensions such as psychoticism, detachment, negative affect, disinhibition, Machiavellianism, and callousness [14, 45]—which would collectively explain substantial variance in both general cognitive ability and social cognition. We hypothesized the p-factor would be negatively associated with social cognition, and

that general cognitive ability would largely drive this association. We further hypothesized most traits related to psychosis and antagonism (e.g., detachment, psychoticism, and callous aggression) would be uniquely negatively associated with social cognition, even after controlling for general cognitive ability, given past work relating these dimensions to worse social cognition [14, 20]. As opposed to these negative associations, we hypothesized Machiavellianism (i.e., dishonesty and manipulativeness) would be associated with better social cognition and general cognitive ability [14]. Finally, we did not have clear hypotheses for associations between social cognition and negative affect or disinhibition but anticipated these factors would be negatively associated with general cognitive ability.

Data for the current study were taken from two separate samples, each of which included measures of psychopathology, social cognition, and general cognitive ability. One sample was from the general population and another sample included patients on the psychosis spectrum and their first-degree biological relatives. Models to evaluate associations of psychopathology dimensions with general and social cognition were computed for each sample. Evaluating associations of psychopathology with social cognition and general cognitive ability in both clinical and general population samples is consistent with dimensional approaches to psychopathology, including HiTOP [11], and can help us understand whether cognitive features of specific psychopathology dimensions are analogous across multiple levels of risk and severity.

## Materials and methods

### Transparency and openness

**Preregistration.** The study was not preregistered.

**Data, materials, code, and online resources.** Data from Sample 1 will be made publicly available on the NIMH Data Archive and through the Connectome Coordination Facility. Data from Sample 2 cannot be made openly available to the public, due to wording in our original institutional review board protocol and consent materials. However, relevant materials, measures, and analytical code are stored on the Open Science Framework. Unrestricted materials will be made publicly available, and readers can request access to private materials (e.g., data from Sample 2) by reaching out to the authors. Our OSF repository can be found at the following link: https://osf.io/d6r2b/

**Reporting.** We report how we determined our sample size, all data exclusions, all manipulations, and all measures in the study.

**Ethical approval.** All studies were approved by the University of Minnesota Twin Cities Institutional Review Board (Sample 1 – 1607M90781 "Neural Disconnection and Errant Visual Perception in Psychotic Psychopathology; Sample 2 – 1002M78152 "Neural Mechanisms of Personality in Decision Making").

### Participants and procedure

Demographic characteristics are reported in Table 1. Specific measures and sample sizes for each sample are summarized in Table 2, and measures are further described in the S1 File. Both datasets had a sample size of more than 300 individuals, which would yield 80% statistical power to detect a correlation of .16 and 90% power to detect a correlation of .19, which are effects in the range anticipated in individual differences research (Gignac & Szodorai, 2016; Richard et al., 2003).

Sample 1—Psychosis Human Connectome Project (P-HCP)—participants (n = 316) were from a large study on the neurobiology of psychosis [46], as part of the series of Connectomes Related to Human Disease funded by the NIH. As all participants who completed the study had above-chance performance on one or more tasks, no participants were excluded based on

**Table 1. Sample demographics.**

| | Samples | |
| --- | --- | --- |
| | **Twin Cities** | **P-HCP** |
| | **(N = 337)** | **(N = 316)** |
| **Demographic data** | | |
| Mean age (SD) | 26.4 (5.1) | 40.3 (13.7) |
| Sex | | |
| Female (%) | 166 (49.3) | 162 (51.3) |
| Male (%) | 171 (50.7) | 153 (48.4) |
| Intersex (%) | | |
| Race/Ethnicity | | |
| White or Caucasian (%) | 246 (73.0) | 240 (75.9) |
| Black, African, or African American (%) | 21 (6.2) | 44 (13.9) |
| Asian or Pacific Islander (%) | 13 (3.9) | 8 (2.5) |
| Latino or Hispanic (%) | 9 (2.7) | 11 (3.5) |
| Multiracial or Other (%) | 44 (13.1) | 11 (3.5) |
| Indigenous, Native American, American Indian, or Alaskan Native (%) | 2 (0.6) | 1 (0.3) |

*Note*. Demographic information was not available for one participant in the P-HCP dataset. Race/ethnicity was assessed in different ways across the two samples. The races and ethnicities listed in the table are an attempt to harmonize these differences. Participants in the Twin Cities sample chose between the following options: Asian or Pacific Islander; Black, African, or African American; Indigenous, Native American, or American Indian; Latino or Hispanic; Multiracial or Other; and White. P-HCP participants chose between Asian or Pacific Islander; American Indian or Alaskan Native; Black, not of Hispanic Origin; Hispanic; Other; and White, not of Hispanic Origin.

**Table 2. Sample size summary.**

| Measure | Twin Cities (n = 337) | P-HCP (n = 316) |
| --- | --- | --- |
| **Psychopathology** | | |
| PID-5 | 330 | 310 |
| ESI | 336 | – |
| SPQ | – | 311 |
| **Social Cognition** | | |
| Mentalizing Vignettes | 331 | – |
| Triangles | – | 251 |
| SAT-MC | – | 219 |
| PROID | – | 126 |
| ER-40 | – | 249 |
| **General Cognitive Ability** | | |
| BACS | – | 310 |
| WRAT Reading | – | 311 |
| WAIS Matrix Reasoning | 335 | 311 |
| WAIS Similarities | 335 | 312 |
| WAIS Picture Vocabulary | 335 | – |
| WAIS Block Design | 335 | – |

*Note*. PID-5 = Personality Inventory for DSM-5, ESI = Externalizing Spectrum Inventory, SPQ = Schizotypal Personality Questionnaire, SAT-MC = Social Attribution Task Multiple Choice, PROID = Performance-based Prosody Identification Test, ER-40 = Penn Emotion Recognition Task, BACS = Brief Assessment of Cognition in Schizophrenia, WRAT = Wide Range Achievement Test, WAIS = Wechsler Adult Intelligence Scale

performance. Participants spanned healthy controls ($n$ = 53), patients with a history of psychosis (schizophrenia = 100, bipolar disorder = 42, schizoaffective disorder-depressive type = 10, schizoaffective disorder-bipolar type = 11, or schizophreniform disorder = 1), and first-degree relatives of these patients ($n$ = 99). Clinical interviews and S1 File were reviewed by a team of at least two doctoral students and/or postdoctoral researchers to determine which diagnostic criteria were met and reach consensus on the most appropriate DSM-IV-TR diagnoses. All participants completed written informed consent and were assessed for their capacity to provide consent [47]. Data collection spanned from 11/21/2016 to 11/04/2021.

Sample 2—Twin Cities—participants (n = 337) were recruited from the community surrounding Minneapolis and St. Paul, MN, primarily through online advertisements, and individuals represented a variety of professions with relatively few students. As all participants who completed the study had above-chance performance on one or more tasks, no participants were excluded based on performance. Data from this sample have been used in multiple published articles, including one paper focused on mentalizing [14], but this article represents the first investigation focusing on how broad and specific psychopathology factors relate to social cognition and general cognitive ability. All participants provided written informed consent. Data collection spanned from 6/25/2011 to 7/31/2013.

**Psychopathology self-report measures.** Dimensional psychopathology measures included the PID-5 [45], Externalizing Spectrum Inventory Brief Form (ESI-BF) [48], and Schizotypal Personality Questionnaire (SPQ) [49]. The PID-5 is a broad measure of maladaptive personality traits spanning antagonism, detachment, disinhibition, negative affect, and psychoticism; the full scale was administered in both samples. The SPQ (Sample 1) and ESI-BF (Sample 2) measure more specific psychopathology dimensions (e.g., facets related to antagonism, psychoticism, and detachment). See the S1 File for additional details on these measures.

**Social cognition measures.** A variety of social cognition measures were administered across the two samples (Table 2), spanning domains of emotion perception, mental state attribution, and mentalizing. Specific measures included the triangle animation task [50], ER-40 [51], POSIT Science Performance-based Prosody Identification Test [52], Social Attribution Task-Multiple Choice [53], and mentalizing vignettes task [54]. The vignettes task included a mentalizing condition and a memory (control) condition. See the S1 File for additional details on these tasks.

**General cognitive ability measures.** To assess general cognitive ability, participants completed subscales of the WAIS-IV [55]. Participants in Sample 1 completed the matrix reasoning and similarities subtests. Sample 2 completed the Block Design, Matrix Reasoning, Vocabulary, and Similarities subtests.

Participants in Sample 1 also completed the Brief Assessment of Cognition in Schizophrenia (BACS) [56] and the word reading subtest of the Wide Range Achievement Test, fourth edition (WRAT-4) [57]. For Sample 2, scores on the memory condition of the mentalizing vignette task were used as an additional indicator of general cognitive ability. See the S1 File for additional details on these tasks.

## Statistical analyses

First-factor saturation across psychopathology scales was assessed with $\omega_h$ and $\omega_t$ (Revelle & Condon, 2019) [58], allowing us to examine the prominence of a p-factor of general psychopathology. Unlike standard implementations of $\omega$, given our broad range of psychopathology measures, computation was done on all scales simultaneously and using scale-scores rather than items. $\omega_t$ represents the amount of shared variance among variables due to both general and specific factors, whereas $\omega_h$ represents the proportion of shared variance among variables

due to a general factor. Subsequently, Velicer's minimum average partial (MAP) [59] tests were used to determine how many factors best explained our observed psychopathology scores in each dataset. Matrix smoothing (via eigenvalue decomposition) was used to account for missing data (Revelle, 2017) [60].

Next, Exploratory Structural Equation Modeling (ESEM) was implemented using *Mplus* [61]. For Sample 1, latent criterion variables were estimated for social cognition and general cognitive ability. Factor indicators for social cognition included accuracy scores on the triangles task (percent accuracy collapsed across conditions), PROID, ER-40, and SAT-MC. Factor indicators for general cognitive ability included scaled-scores on the WAIS subtests for matrix reasoning and similarities, BACS standardized composite scores, and standardized scores on the WRAT reading test; composite scores on the BACS were used due to the large number of individual tests within the BACS and to balance the number of indicators across social cognition and general cognitive ability.

In Sample 2, criterion variables included observed scores for the mentalizing vignettes task (mentalizing condition) and a latent variable for general cognitive ability. Indicators of general cognitive ability included scaled scores on the four WAIS subtests, as well as scores on the memory condition from the mentalizing vignettes task. Residual covariance for the two mentalizing vignettes conditions was freely estimated.

For each sample, exploratory factors were estimated using the psychopathology subscales. Given our interest in a general factor of psychopathology—and its association with social cognition and general cognitive ability—we used an exploratory bi-factor rotation (oblique bi-geomin with ε = .1). Although many studies involving the p-factor have relied on confirmatory bifactor models, exploratory models that include cross-loadings among specific factors allow for more accurate estimation of both general and specific factors [62, 63]. Our models were estimated using the number of factors indicated by Velicer's MAP tests plus a general factor. Each specific factor was allowed to correlate with other specific factors but was orthogonal to the general factor. Method factors were also estimated to model shared variance of subscales from the same questionnaire (i.e., two factors, each indicated by subscales of the PID-5 and SPQ or ESI, respectively), which resulted in significantly better fit versus models without these methods factors. The method factors were set to be orthogonal to the exploratory psychopathology variables and were not used as statistical predictors of social cognition or general cognitive ability. Results were similar if methods factors were not included.

Moving on to the structural model, the exploratory psychopathology factors were used as simultaneous statistical predictors of social cognition and general cognitive ability. Moreover, we used a path model to decompose associations of psychopathology factors with social cognition (i.e., total associations) into associations independent of general cognitive ability (i.e., direct associations) and associations that could be accounted for by general cognitive ability (i.e., indirect associations). Full information maximum likelihood (FIML) estimation was used.

Subsequently, two sets of follow-up analyses were conducted for each sample. First, we conducted post-hoc, exploratory moderation analyses to examine whether any psychopathology variables significantly interacted with general cognitive ability, in statistically predicting social cognition. To do so, we extracted estimated factor scores for psychopathology dimensions and general cognitive ability (as well as for Social Cognition in the P-HCP dataset) using the regression method in *Mplus*. Interaction terms were then computed for each psychopathology dimension by multiplying corresponding factor-score estimates with the factor-score estimates for general cognitive ability. Then, we conducted regression analyses statistically predicting social cognition using general cognitive ability, each psychopathology factor score, and each of the general cognitive ability-by-psychopathology interaction terms.

**Table 3. Tests of factor structure.**

| Dataset | $\omega_h$ | $\omega_t$ | MAP Factors | MAP Value |
|---|---|---|---|---|
| P-HCP | .75 | .97 | 5 | .018 |
| Twin Cities | .69 | .96 | 7 | .014 |

Finally, we conducted post-hoc regression analyses using these extracted factor scores to confirm whether key findings remained significant when controlling for sex and to examine whether there were any significant psychopathology-by-sex interactions in statistically predicting social cognition. In these analyses, sex was included as a dummy-coded predictor variable (females = 0, males = 1). Interaction terms were then computed for each psychopathology dimension by multiplying corresponding factor-score estimates by the sex dummy variable. Then, we conducted regression analyses statistically predicting social cognition using general cognitive ability, sex, each psychopathology factor score, and each of the sex-by-psychopathology interaction terms. Simple-slopes tests were then done to further examine any significant psychopathology-by-sex interactions.

## Results

In terms of first-factor saturation, values of $\omega_h$ were .75 for Sample 1 and .69 for Sample 2 (Table 3), suggesting a relatively strong p-factor in each sample. For Sample 1, Velicer's MAP test suggested five factors, whereas seven factors were suggested in Sample 2 (Table 3). For subsequent analyses, we used solutions based on the number of factors suggested by MAP plus an additional general factor, which resulted in the most interpretable models.

ESEM fit indices showed good fit, for both datasets (Table 4). For social cognition and general-cognitive-ability latent variables, all indicators loaded significantly onto their latent variable (Fig 1). Standardized loadings for psychopathology factors are displayed in Table 5 for Sample 1 and Table 6 for Sample 2. Each sample showed a strong p-factor with various specific psychopathology factors. Each sample also had a clear Machiavellianism factor, with strong loadings for measures of manipulativeness and deceitfulness. The content of other factors varied slightly across the two datasets (likely as a function of including the SPQ in Sample 1 versus ESI-BF in Sample 2). For instance, in Sample 1, there were separate eccentricity and psychoticism factors, whereas these formed a single factor in Sample 2. Sample 2 had a clear callous-aggression factor, whereas indicators of this factor loaded negatively onto a factor we labeled emotionality in Sample 1; this emotionality factor also had positive loadings for anxiety-related variables, which loaded onto detachment in Sample 2. Detachment factors had indicators that loaded in opposite directions for Samples 1 versus 2 (so we reversed this factor in Sample 2 to aid with interpretation). Finally, Sample 2 showed additional factors not represented in Sample 1, which we labeled suspiciousness, perfectionism, and dependability.

Several psychopathology factors were significantly associated with general cognitive ability, in each of the datasets (Tables 7 and 8). In both datasets, detachment was negatively related to social cognition, even after controlling for variance in general cognitive ability; this suggests that detachment—a construct related to social isolation and intimacy avoidance—shows

**Table 4. Model fit statistics.**

| Models | RMSEA | 95% C.I. | $p$ | SRMR | CFI |
|---|---|---|---|---|---|
| P-HCP | .057 | [.052, .061] | .010 | .041 | .930 |
| Twin Cities | .053 | [.049, .057] | .123 | .032 | .920 |

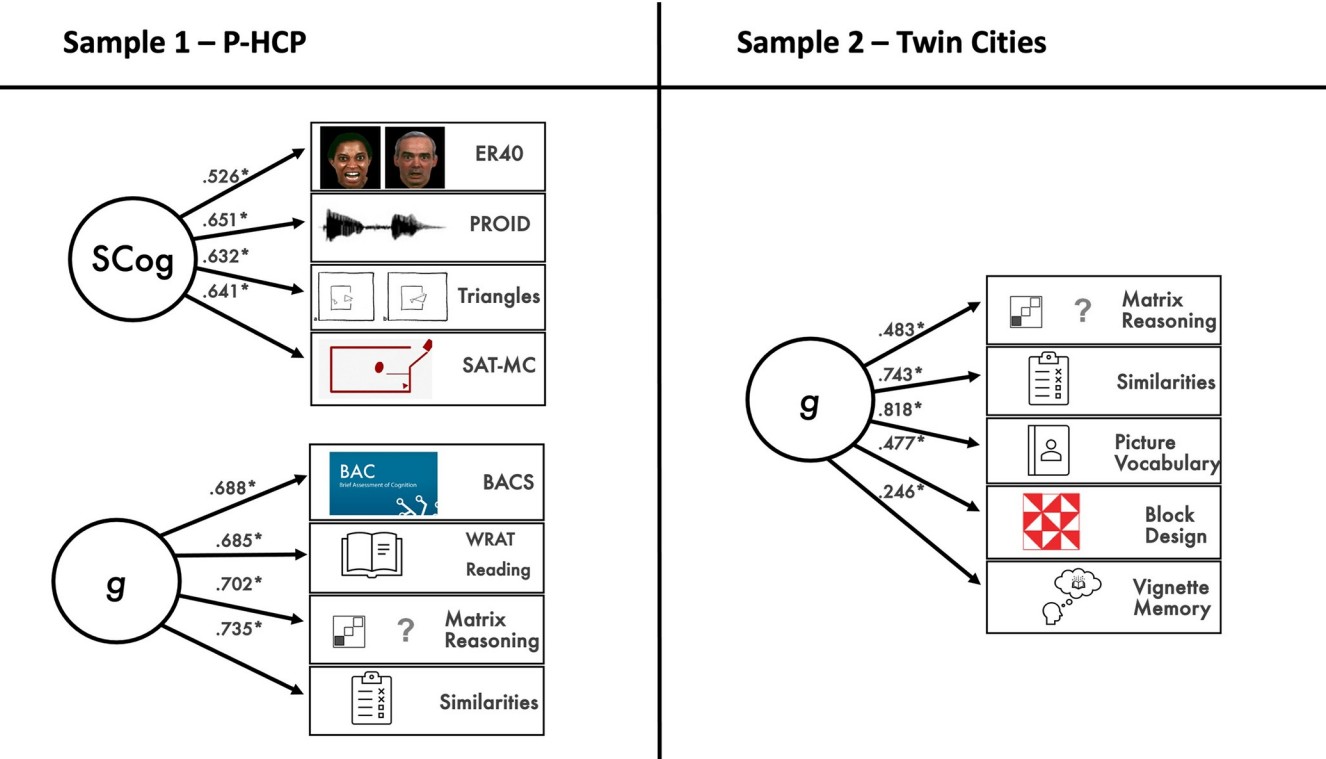

**Fig 1. Measurement models for social and general cognitive ability.** *Note.* *$p$ < .05. Standardized measurement models for social cognition and general cognitive ability.

independent associations with social cognition, not explained by covariance with general cognitive ability. In both datasets, Machiavellianism and Eccentricity (which combined with psychoticism in Sample 2) showed positive associations with general cognitive ability and positive total associations with social cognition. In Sample 1, emotionality showed a positive association with general cognitive ability and positive indirect association with social cognition; a similar pattern was seen for detachment in Sample 2, which included overlapping indicators with Sample 1's emotionality factor (e.g., PID-5 anxiety). In Sample 1, the p-factor showed a negative association with general cognitive ability and a negative indirect association with social cognition; this indicates that any association of the p-factor with social cognition was nearly fully accounted for by general cognitive ability. However, no associations of the p-factor and social cognition were found in Sample 2. Contrary to our hypotheses, the callous-aggression factor in Sample 2 was not significantly associated with social cognition or general cognitive ability.

In Sample 1, there were no significant interaction effects of psychopathology and general cognitive ability when statistically predicting social cognition. In Sample 2, there was a significant negative interaction between general cognitive ability and dependability in statistically predicting social cognition ($\beta$ = -.108, $p$ = .019); given that both general cognitive ability and dependability positively predicted social cognition, a negative interaction indicates an attenuation of these positive effects given increases in the other respective variable.

In both samples, all effects identified as significant in our primary ESEM analyses remained significant when controlling for sex (and directions of effects remained consistent). When accounting for general cognitive ability and psychopathology, Females showed better social cognition than males in both Samples 1 ($\beta$ = -.066, $p$ = .008) and 2 ($\beta$ = -.104, $p$ = .034).

**Table 5. Factor loadings for P-HCP.**

| Questionnaire Variable | p-factor | Detachment | Psychoticism | Machiavellian | Eccentricity | Emotionality |
|---|---|---|---|---|---|---|
| PID Withdrawal | .555* | **.651***  | -.048 | -.066 | -.070 | .048 |
| SPQ No Close Friends | .556* | **.607*** | -.020 | -.056 | -.100* | -.033 |
| PID Restricted Affect | .358* | **.512*** | .079 | .022 | .189* | -.021 |
| PID Anhedonia | .704* | **.453*** | -.105 | .116 | -.017 | .059 |
| SPQ Constricted Affect | .616* | **.454*** | -.053 | -.104* | .103* | -.010 |
| PID Intimacy Avoidance | .383* | **.335*** | .054 | -.196* | -.128* | -.062 |
| PID Emotional Lability | .720* | **-.250*** | -.047 | .003 | -.061 | .179* |
| SPQ Social Anxiety | .590* | **.245*** | -.077 | -.186* | -.111* | .126* |
| PID Separation Insecurity | .545* | **-.220*** | -.061 | .166* | -.148* | .214* |
| PID Depressivity | .827* | **.190*** | -.159 | .170 | -.060 | .147 |
| PID Unusual Beliefs | .750* | .036 | **.563*** | .024 | .026 | .018 |
| SPQ Odd Beliefs | .535* | -.076* | **.481*** | .080 | -.034 | -.040 |
| SPQ Ideas of Reference | .692* | -.179* | **.248*** | -.128* | -.127* | .014 |
| PID Perception | .839* | -.039 | **.227*** | -.055 | .070* | .002 |
| PID Hostility | .632* | -.051 | **-.203*** | .103 | -.045 | -.068 |
| SPQ Unusual Perception | .740* | -.069* | **.322*** | -.120* | .005 | -.103* |
| PID Manipulativeness | .271* | -.068 | .002 | **.572*** | -.001 | -.004 |
| PID Deceitfulness | .523* | .015 | -.090* | **.514*** | -.002 | -.084* |
| PID Attention Seeking | .350* | -.370* | .033 | **.422*** | .120* | .003 |
| PID Grandiosity | .355* | .043 | .176* | **.335*** | -.001 | .010 |
| PID Distractibility | .786* | -.070 | -.116* | **-.176*** | .155* | -.002 |
| SPQ Suspiciousness | .727* | .001 | .092 | -.110* | **-.372*** | -.100* |
| PID Suspiciousness | .750* | .053 | .118* | .027 | **-.340*** | -.065 |
| PID Eccentric | .806* | .015 | .141* | .033 | **.337*** | -.018 |
| SPQ Eccentric | .680* | .010 | .095 | .002 | **.319*** | -.071 |
| SPQ Odd Speech | .735* | -.060 | -.017 | -.147* | **.223*** | -.024 |
| PID Submissiveness | .280* | -.011 | -.092 | .102 | .064 | **.488*** |
| PID Irresponsibility | .647* | -.055 | -.134* | .077 | -.109* | **-.370*** |
| PID Anxiety | .765* | -.068 | -.114* | -.036 | -.129* | **.352*** |
| PID Impulsivity | .654* | -.243* | -.161* | -.022 | .035 | **-.311*** |
| PID Callousness | .489* | .217* | -.035 | .194* | -.019 | **-.305*** |
| PID Risk Taking | .516* | -.041 | -.024 | -.105 | .083 | **.275*** |
| PID Perfectionism | .812* | -.021 | -.094* | -.063 | .171* | **.211*** |
| PID Perseveration | .555* | .651* | -.048 | -.066 | -.070 | .048 |

Note

* $p < .05$. Highest loadings (other than the p-factor) for each observed variable are noted using bold. SPQ = Schizotypal Personality Questionnaire, PID = Personality Inventory for DSM-5, ESI = Externalizing Spectrum Inventory

In Sample 1, there was a significant, negative interaction between sex and Emotionality ($\beta$ = -.053, $p$ = .035). Given that females performed better on social cognition and higher Emotionality was marginally associated with worse social cognition, this negative interaction appears to indicate that the association between Emotionality and worse social cognition may be particularly prominent in males. Indeed, follow-up simple-slope tests suggest that males show a fairly strong association between Emotionality and worse social cognition ($\beta$ = -.284, $p$ < .001), females show a weaker (but still significant) negative association ($\beta$ = -.091, $p$ = .027).

In Sample 2, there was a significant, positive interaction between sex and Detachment ($\beta$ = .128, $p$ = .008). Given that females performed better on social cognition and that higher

**Table 6. Factor loadings for Twin Cities.**

| Questionnaire Variable | p-factor | Eccentric | Machiavellian | Detach | Callous-Agg | Suspicious | Perfect | Depend |
|---|---|---|---|---|---|---|---|---|
| PID Unusual Beliefs | .396* | **.569*** | .066 | -.119 | -.100 | .049 | .047 | -.229 |
| PID Eccentric | .576* | **.470*** | -.069 | -.001 | -.020 | -.144 | .019 | .149 |
| PID Restricted Affect | .433* | **.468*** | -.015 | .134 | .223 | .079 | .029 | .238 |
| PID Perception | .560* | **.402*** | -.005 | -.007 | -.201* | .003 | -.011 | -.042 |
| PID Emotional Lability | .486* | **-.251*** | -.018 | .032 | -.269 | .036 | .150 | .049 |
| PID Deceitfulness | .619* | .059 | **.648*** | .064 | .093 | -.044 | -.057 | -.047 |
| PID Manipulativeness | .483* | -.009 | **.552*** | -.182* | -.041 | -.024 | .221 | -.063 |
| ESI Honesty | -.488* | .075 | **-.472*** | -.025 | -.153 | .008 | .332* | -.024 |
| PID Grandiosity | .336* | .161* | **.344*** | -.064 | .111 | -.028 | .320* | -.005 |
| PID Separation Insecurity | .360* | -.191 | **.198*** | .040 | -.171 | .076 | -.051 | .170 |
| PID Risk Taking | .304* | .173 | -.049 | **-.659*** | .106 | -.011 | -.066 | .024 |
| PID Anhedonia | .613* | .045 | -.156* | **.528*** | .097 | -.026 | -.133 | .028 |
| PID Impulsivity | .743* | -.059 | -.068 | **-.397*** | -.173* | -.033 | -.181 | .045 |
| PID Withdrawal | .565* | .248 | -.114 | **.360*** | .184* | .073 | -.018 | .110 |
| PID Depressivity | .674* | -.044 | -.057 | **.359*** | -.105 | .048 | -.183* | .083 |
| PID Attention Seeking | .373* | -.021 | .264* | **-.316*** | -.188 | -.140 | .217 | .032 |
| PID Anxiety | .570* | -.199 | -.026 | **.297*** | -.201* | .150 | .016 | .123 |
| ESI Rebelliousness | .605* | .132* | -.101 | **-.288*** | .153* | -.066 | -.086 | -.045 |
| ESI Impulsivity | .710* | -.179* | -.155* | **-.272*** | -.018 | -.046 | -.117 | -.134 |
| ESI Impatience | .533* | -.105 | .000 | **-.210*** | .013 | -.034 | .103 | .076 |
| PID Intimacy Avoidance | .524* | .192 | -.031 | **.144*** | -.090 | .111 | .020 | .076 |
| PID Callousness | .569* | -.014 | .131* | .036 | **.703*** | .040 | .049 | .008 |
| ESI Empathy | -.362* | -.037 | -.059 | -.080 | **-.684*** | -.038 | .105 | -.146 |
| ESI Physical Aggression | .377* | -.139 | -.181* | -.178* | **.451*** | .030 | .177 | -.059 |
| ESI Destructive Aggression | .424* | -.030 | .021 | -.079 | **.314*** | -.139 | .129 | -.121 |
| PID Submissiveness | .083 | -.116 | .269* | .126* | **-.271*** | -.175 | .011 | .266 |
| ESI Theft | .413* | .070 | .070 | -.123 | **.224*** | .016 | -.100 | -.081 |
| PID Perseveration | .693* | .073 | -.014 | .023 | **-.194*** | -.044 | .115 | .161* |
| PID Suspiciousness | .594* | .016 | -.026 | .074 | .058 | **.572*** | .038 | -.028 |
| ESI Alienation | .570* | .000 | -.121* | .022 | -.117* | **.455*** | .066 | -.080 |
| ESI Blame Externalization | .526* | -.044 | -.005 | -.083 | .040 | **.299*** | -.041 | .142* |
| PID Distractibility | .711* | .066 | -.115* | .032 | -.246* | **-.268*** | .010 | .092 |
| ESI Relational Aggression | .618* | -.098 | .076 | .025 | .320 | **-.220*** | .223 | -.107 |
| PID Perfectionism | .244* | .034 | .026 | -.008 | -.112 | .177* | **.426*** | .133 |
| PID Hostility | .666* | -.233* | -.028 | .074 | .177 | .024 | **.358*** | .046 |
| ESI Planfulness | .652* | .156* | .010 | .164 | .070 | .087 | **.368*** | .030 |
| ESI Irresponsibility | .605* | .023 | -.043 | -.006 | .027 | .030 | .004 | **-.422*** |
| PID Irresponsibility | .769* | .043 | .106* | .158* | -.147* | -.012 | -.092 | **-.391*** |
| ESI Dependability | -.587* | .057 | .002 | -.141* | .124 | .204* | .128 | **.326*** |
| ESI Fraud | .572* | -.037 | .122* | -.075 | .125 | .130 | .030 | **-.287*** |
| ESI Boredom Proneness | .568* | -.013 | -.076 | .051 | .047 | -.024 | .009 | **.169*** |
| ESI Excitement Seeking | .035 | -.061 | -.023 | -.024 | -.055 | -.009 | .045 | .095 |

Note

* $p < .05$. Highest loadings (other than the p-factor) for each observed variable are noted using bold. SPQ = Schizotypal Personality Questionnaire, PID = Personality Inventory for DSM-5, ESI = Externalizing Spectrum Inventory

**Table 7. Associations for P-HCP.**

| Psychopathology Factor | Total association with SCog | Direct association with SCog | Indirect association with SCog | Association with GCA |
|---|---|---|---|---|
| p-factor | -.132 (p = .082) | .141 (p = .840) | **-.273 (p < .001)** | **-.299 (p < .001)** |
| Detachment | -.080 (p = .343) | **-.174 (p = .028)** | .094 (p = .187) | .103 (p = .169) |
| Psychoticism | **-.175 (p = .045)** | -.098 (p = .221) | -.077 (p = .293) | -.084 (p = .290) |
| Machiavellianism | **.196 (p = .032)** | -.086 (p = .387) | **.282 (p = .001)** | **.309 (p < .001)** |
| Eccentric | **.318 (p < .001)** | -.095 (p = .382) | **.413 (p < .001)** | **.452 (p < .001)** |
| Emotionality | .147 (p = .116) | -.207 (p = .062) | **.354 (p < .001)** | **.387 (p < .001)** |

*Note.* Significant associations are bolded. SCog = social cognition, GCA = general cognitive ability.

Detachment was associated with worse social cognition, this positive interaction appears to indicate that the association between Detachment and worse social cognition may be particularly prominent in females. Indeed, follow-up simple-slope tests suggest that females showed a significant association between Detachment and worse social cognition ($\beta$ = -.247, $p < .001$), whereas the association was not significant in males ($\beta$ = -.026, $p = .694$).

## Discussion

We sought to investigate how both general and specific psychopathology dimensions were associated with social cognition, and whether these associations could be accounted for by general cognitive ability. A multi-sample approach was used to examine whether associations were consistent across multiple levels of psychopathology risk/severity. In both samples, detachment was associated with worse social cognition, even when controlling for general cognitive ability. In our sample that included psychosis patients and their first-degree relatives,

**Table 8. Associations for Twin Cities.**

| Psychopathology Factor | Total association with SCog | Direct association with SCog | Indirect association with SCog | Association with GCA |
|---|---|---|---|---|
| p-factor | .012 (p = .841) | .053 (p = .318) | -.042 (p = .227) | -.083 (p = .214) |
| Eccentricity | **.228 (p = .025)** | .100 (p = .224) | **.128 (p = .016)** | **.253 (p = .008)** |
| Machiavellian | **.171 (p = .006)** | .096 (p = .115) | **.075 (p = .050)** | **.148 (p = .044)** |
| Detachment | .087 (p = .241) | **-.250 (p = .001)** | **.163 (p = .002)** | **.323 (p < .001)** |
| Callous Aggression | -.197 (p = .149) | -.157 (p = .314) | -.040 (p = .445) | -.080 (p = .445) |
| Suspiciousness | -.065 (p = .445) | **.207 (p = .031)** | **-.272 (p < .001)** | **-.539 (p < .001)** |
| Perfectionism | -.064 (p = .490) | -.071 (p = .443) | .007 (p = .891) | .014 (p = .891) |
| Dependability | **.226 (p = .013)** | **.178 (p = .039)** | .049 (p = .362) | .096 (p = .367) |

*Note.* Significant associations are bolded. SCog = social cognition, GCA = general cognitive ability.

general psychopathology was negatively associated with social cognition, with general cognitive ability accounting for much of this association. Conversely, Machiavellianism and eccentricity were positively associated with both social cognition and general cognitive ability. The fact that several findings generalized across samples suggests these associations are robust. This work fills important gaps in our understanding of the long-studied associations between social cognition and various psychopathology dimensions, approaching the topic from a hierarchical, dimensional perspective and clarifying the role of general cognitive ability. In addition to these primary contributions, our findings also echo previous work that shows superior social cognitive ability in females [31, 54, 64] and offer preliminary findings to inform future research on how sex and psychopathology may interact to explain variance in social cognition.

The negative association between p-factor and general cognitive ability in Sample 1 is unsurprising and replicates previous research [10, 36, 37]. However, to our knowledge, this is the first study to show a negative association of p with social cognition and to suggest this relation is explained largely by variance shared with general cognitive ability. Although it is possible that social cognitive impairment is truly a transdiagnostic feature associated with broad risk for psychopathology, much of the impairment in social cognition seen across various psychopathology dimensions may be a function of general cognitive deficits rather than problems specific to social processes. These findings suggest it may be important for researchers to control for the effects of general cognitive ability (e.g., by including an effective control task or statistically controlling for general cognitive ability) when investigating social cognition and its relation to psychopathology. Contrary to our hypotheses, however, the p-factor was not associated with social cognition or general cognitive ability in Sample 1; this may be due to range restriction for psychopathology that occurs when drawing participants from the general population (as in Sample 1) versus specifically recruiting patients and their relatives (as in Sample 2).

In contrast to the p-factor, detachment was incrementally related to social cognitive deficits even after controlling for general cognitive ability. Detachment is a dimension of psychopathology that is very clearly related to maladaptive social functioning, spanning deficits in prosociality and desire to bond with others. Findings of a specific negative relation between detachment (or negative schizotypy) and social cognition are in line with both the theoretical basis and content of this factor [49] and empirical findings that, of the schizotypy subfactors, only negative schizotypy was uniquely associated with interpersonal functioning [20]. Similarly, among those with schizophrenia and other psychotic disorders, negative symptoms are typically the stronger predictor of social cognition and social functioning [65–69]. It is additionally interesting to consider the detachment findings in the context of the different pattern of findings for psychoticism and suspiciousness, which suggests that dysfunction in social cognition may be more proximally related to detachment but driven by lower general cognitive ability for the other two dimensions. Future research should more closely examine whether these patterns also hold across the psychosis spectrum when using more traditional measures of positive versus negative symptoms.

Whereas most psychopathology dimensions showed a negative association with general and social cognition, there were a few exceptions. Both Machiavellianism and eccentricity showed positive indirect associations with social cognition and positive associations with general cognitive ability. Our finding of a positive association between social cognition and a factor representing dishonesty, deceit, and manipulativeness (i.e., Machiavellianism) replicates work finding a similar association using a single mentalizing task [14] and more recent working using multiple tasks [70]. This also further supports the longstanding notion that being able to successfully persuade, manipulate, and deceive can be facilitated by understanding the thoughts and emotions of others [71]. Taken with previous work and theory, current findings

seem to suggest that specific antagonistic traits may be associated with enhanced abilities in some contexts. Eccentricity was also positively related to general and social cognition, which is in line with past research indicating a positive association of openness to experience—a trait theoretically and empirically linked to unconventionality and eccentricity [72]—with empathy and other measures of interpersonal functioning [73]. The positive association between our eccentricity factor and general cognitive ability is also consistent with broader research showing positive associations between openness and intelligence [74]. Finally, the diverging associations of social cognition with psychoticism versus eccentricity—two highly related traits—in Sample 1 is also in line with research and theory that poses these constructs as part of a continuum with openness and intellect, in which strength of association with general cognitive ability varies systematically along that continuum [32, 33, 72].

Finally, it is worth highlighting our finding that factors marked by negative affect (emotionality in Sample 1 and detachment in Sample 2) showed positive associations with general cognitive ability and positive indirect associations with social cognition. No distinct negative affect factor emerged in either dataset, which makes interpretation of these findings more difficult. However, this is not particularly surprising, as negative affect (and the related construct *neuroticism*) is a broad risk factor for all psychopathology, and its variance overlaps substantially with the p-factor [75, 76]. However, our specific finding is in line with a small but growing body of research suggesting similar positive associations between cognitive performance and negative affect or mood disorders [77, 78] and with the previous finding of a positive association between neuroticism and social cognition [79]. Findings are also in line with at least some theoretical perspectives. For instance, the theory of depressive realism—which has garnered mixed empirical support—posits that individuals who are depressed may have a more realistic perception of certain aspects of reality [80–82]. Indeed, positive associations could arise from an increased awareness of emotional or general sensory information that may contribute to experience of negative affect. It may also be that tendencies toward negative affect may be more specifically associated with cognitive abilities, after accounting for the p-factor (which likely captures variance related to demoralization and general distress).

In addition to showing associations with social cognition, in Sample 1, negative affect variables formed a factor that also included low callousness/high empathy. This is consistent with the notion that negative affect may be associated with prosociality, at least among some individuals, and is in line with research from the HEXACO perspective on personality, which posits a core personality trait of emotionality, combining features of negative affect with sympathy and compassion [83, 84]; the emergence of this factor is also in line with the notion of callous-unemotional traits as a construct [85]. Altogether, our results provide tentative evidence that negative affect may be associated with enhanced social cognition. Specific mechanisms of these possible associations between (social) cognitive abilities and stronger emotionality or negative affect warrant further investigation.

## Limitations

Generalizability across samples and in both clinical and general population contexts is a strength of the current study. Nonetheless, some discrepancies were observed across datasets and their sources warrant discussion. For one, associations could be influenced by sample differences, including the nature of the populations (patients, relatives, and controls versus community members). Moreover, variations in the psychopathology scales administered across the samples likely led to factors that differ slightly in their content and, thus, their associations with task performance. Perhaps the most likely contributor to inconsistent associations is the differences in social cognition measures across the datasets. For example, the Twin Cities

dataset used only a single social cognition task, weighted heavily toward language processing and higher-order mentalizing, whereas the P-HCP dataset used a battery focused on perception and mental state attribution. Given these limitations, our findings should be interpreted as preliminary. Future research should be conducted to confirm the present results in a large sample with data designed specifically to assess relations among psychopathology dimensions, general cognitive ability, and social cognitive ability (across multiple domains in the same participants). This would allow us to determine whether the present findings are robust and generalizable.

Some other limitations should be addressed in future work. First, our patient sub-sample focused on individuals with psychosis (though the relatives in the P-HCP sample had mood and anxiety disorders at rates higher than typically seen in the general population); future work could sample patients across a wider range of diagnoses associated with social dysfunction—including personality disorders and social anxiety—to capture a broader range of elevated antagonism and negative affect. Second, our samples did not all include reports of income-level, geographic background, or other relevant socioeconomic information, which is a potential confounding factor that should be thoroughly examined and/or controlled for in future work. Finally, the current study focused on social cognitive ability from the perspective of overall accuracy, but other components (e.g., parsing errors into hypermentalizing versus hypomentalizing, distinguishing lack of perceptual sensitivity from random or careless responding or examining response bias toward specific emotions) may be just as important in understanding how psychopathology is related to social functioning. This issue could be addressed in future work using theoretical frameworks and/or computational models that decompose task performance into cognitive components, such as evidence evaluation efficiency, social learning, prior beliefs, and perceptual biases [33, 86, 87].

## Conclusions

To our knowledge, this study is the first to examine associations between social cognition and psychopathology using a hierarchical, dimensional approach. This research harnessed the utility of latent variables, incorporated a broad range of tasks, and used measures spanning many of the psychopathology dimensions relevant to social functioning. We provide evidence that detachment (negative schizotypy) was associated with worse social cognition, even after controlling for general cognitive ability, whereas associations of social cognition with general psychopathology could be accounted for by general cognitive ability. These findings suggest that some psychopathology dimensions (i.e., those most relevant to social withdrawal) may be proximally related to social cognitive deficits, whereas other traits are more distally related to social cognition, via broader cognitive pathways. In contrast, eccentricity and Machiavellianism were positively associated with social cognition and general cognitive ability, indicating that certain psychopathology dimensions may, somewhat surprisingly, be related to cognitive advantages.

Adopting a dimensional, hierarchical approach to studying mechanisms and correlates of psychopathology can be a useful tool. Such work may have the power to inform future translational research and pathways to personalized medicine, as interventions for social dysfunction may benefit from targeting specific psychopathology dimensions and associated mechanisms (e.g., general cognitive ability versus processes specific to social cognition). Future work incorporating computational models, neuroimaging, and real-world experience sampling would help further elucidate broad and specific mechanisms of social functioning as related to psychopathology.

### Prior versions

Preliminary analyses were presented by S.D.B. at conferences hosted by the Society for Interpersonal Theory and Research and Society for Research in Psychopathology. This work was posted online as a preprint: https://psyarxiv.com/gqaju/.

## Supporting information

**S1 File. Supplemental information.** This file contains supporting methodological information.
(PDF)

## Author Contributions

**Conceptualization:** Scott D. Blain, Jerillyn S. Kent, Timothy A. Allen, Carly A. Lasagna, Aisha L. Udochi, Scott R. Sponheim, Colin G. DeYoung, Ivy F. Tso.

**Data curation:** Scott D. Blain, Jerillyn S. Kent, Timothy A. Allen, Chloe A. Peyromaure de Bord, Aisha L. Udochi.

**Formal analysis:** Scott D. Blain.

**Funding acquisition:** Scott R. Sponheim, Colin G. DeYoung, Ivy F. Tso.

**Investigation:** Scott D. Blain, Scott R. Sponheim, Colin G. DeYoung, Ivy F. Tso.

**Methodology:** Scott D. Blain, Timothy A. Allen, Chloe A. Peyromaure de Bord, Aisha L. Udochi, Scott R. Sponheim.

**Project administration:** Scott D. Blain, Scott R. Sponheim, Colin G. DeYoung, Ivy F. Tso.

**Resources:** Scott R. Sponheim, Colin G. DeYoung, Ivy F. Tso.

**Software:** Scott R. Sponheim, Colin G. DeYoung, Ivy F. Tso.

**Supervision:** Scott R. Sponheim, Colin G. DeYoung, Ivy F. Tso.

**Validation:** Scott D. Blain.

**Visualization:** Scott D. Blain, Carly A. Lasagna.

**Writing – original draft:** Scott D. Blain.

**Writing – review & editing:** Scott D. Blain, Jerillyn S. Kent, Timothy A. Allen, Carly A. Lasagna, Chloe A. Peyromaure de Bord, Aisha L. Udochi, Scott R. Sponheim, Colin G. DeYoung, Ivy F. Tso.

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
