## [Decision Letter · Decision Letter 0]

1 Oct 2024

PONE-D-24-24635Constructs Across a Hierarchical, Dimensional Model of Psychopathology Show Differential Associations with Social and General Cognitive AbilityPLOS ONE

Dear Dr. Blain,

Thank you for submitting your manuscript to PLOS ONE. After careful consideration, we feel that it has merit but does not fully meet PLOS ONE’s publication criteria as it currently stands. Therefore, we invite you to submit a revised version of the manuscript that addresses the points raised during the review process.

Be sure to:Indicate which changes you require for acceptance versus which changes you recommendAddress any conflicts between the reviews so that it's clear which advice the authors should followProvide specific feedback from your evaluation of the manuscriptPlease ensure that your decision is justified on PLOS ONE’s publication criteria and not, for example, on novelty or perceived impact.

We look forward to receiving your revised manuscript.

Kind regards,

Soo-Hee Choi

Academic Editor

PLOS ONE

Journal Requirements:

2. Thank you for stating the following financial disclosure: This research was supported by the National Institute of Health (U01MH108150 to S.R.S., R03DA029177-01A1 to C.G.D., R01MH122491-01 to I.F.T.) and National Science Foundation (SES-1061817 to C.G.D. and Graduate Research Fellowships to S.D.B., C.A.L, and A.L.U.).  

Reviewers' comments:

Reviewer's Responses to Questions

**Comments to the Author**

1. Is the manuscript technically sound, and do the data support the conclusions?

Reviewer #1: Partly

Reviewer #2: Yes

2. Has the statistical analysis been performed appropriately and rigorously? 

Reviewer #1: Yes

Reviewer #2: Yes

3. Have the authors made all data underlying the findings in their manuscript fully available?

Reviewer #1: No

Reviewer #2: No

4. Is the manuscript presented in an intelligible fashion and written in standard English?

Reviewer #1: Yes

Reviewer #2: Yes

5. Review Comments to the Author

Reviewer #1: In this well-performed and interesting study the authors examine dimensions of psychopathology and associations with social and general cognitive ability.

The study is relevant in exploring the dimensional approach to psychopatholgy using the HiTOP model, and the analyses are golden standard. There are some minor considerations, mainly pertaining to data quality and presentation of results:

1. The authors mention themselves in the limiation section the important issue with the quality of data, such as the use of only one social cognitive measure in Sample 1 (Mentalizing Vignettes), which comprises other subdomains than those used in Sample 2; the use of self-seport questionnaries to measure dmensions of psychopathology; and recruting related issues / sample diffences (sign. different age e.g.). The study appear with a pragmatic use of existing data pooled from other original studies, and must be very clear in this explorative basis. Although hypothesis are introduced, the datacollection were not designed for the current study. Following, any results and conclusions must be clearly stated cautious, not generalizable and only hypothesis generating for further examination. Furthermore, the difference in social cognitive measures may not only contribute to inconsistent associations as stated - but also consistent associations may be a random effect, as we cannot be sure the same concepts are measured.

2. I do miss an attempt to clarify in post hoc testing the more specific dynamic effect of GCA in the relation between p-factor and social cognition. Is it potentially mediating or moderating the effect?

3. The should be seperate analyses for sex, as this may be important for both psychopathology and social cognition.

A few minor comments:

- Rhetorical questions in an introduction is a really bad style.

- ASD is not a psychiatric disorder, but a neurodevelopmental disorder. The distinction is important for ASD subjects.

Reviewer #2: PONE-D-24-24635

Constructs Across a Hierarchical, Dimensional Model of Psychopathology Show Differential Associations with Social and General Cognitive Ability

EVALUATION

This is really interesting research addressing the relationships between social cognition and psychopathology studying two samples (Psychosis Human Connectome Project, P-HCP, and Twin cities –community sample).

Importantly, the role of cognitive ability (aka, intelligence) is also considered.

Furthermore, a latent variable approach is applied here.

One of the key findings is that the relationship between social cognition and the p factor can be accounted for by cognitive ability, while detachment is related with social cognition even after controlling for cognitive ability.

Tables 7 and 8 depict important findings that deserve attention.

Comparison among values obtained from both samples are highly informative. Thus, for instance, the -.30 value between general cognitive ability and the p factor in the P-HCP samples is in high contrast with the null value for this relationship in the Twin cities sample.

The general picture provides important information.

The Ms. is well written and goes to the heart of the main research questions.

Therefore, I recommend publication.

6. PLOS authors have the option to publish the peer review history of their article (what does this mean?). If published, this will include your full peer review and any attached files.

Reviewer #1: No

Reviewer #2: No

---

## [Author Response · Author response to Decision Letter 0]

1 Dec 2024

We thank the editor and reviewers for their thoughtful responses to our paper and for their overall positive reception of our work. We have revised our paper as follows, responding to each of the relevant critiques. Revisions are also noted in the main text, using red. We appreciate PLOS One’s continued considered of our work and look forward to the opportunity to publish our research in the journal.

Reviewer #1: In this well-performed and interesting study, the authors examine dimensions of psychopathology and associations with social and general cognitive ability.

The study is relevant in exploring the dimensional approach to psychopathology using the HiTOP model, and the analyses are golden standard. There are some minor considerations, mainly pertaining to data quality and presentation of results:

1. The authors mention themselves in the limitation section the important issue with the quality of data, such as the use of only one social cognitive measure in Sample 1 (Mentalizing Vignettes), which comprises other subdomains than those used in Sample 2; the use of self-report questionaries to measure dimensions of psychopathology; and recruiting related issues / sample differences (sign. different age e.g.). The study appear with a pragmatic use of existing data pooled from other original studies, and must be very clear in this explorative basis. Although hypothesis are introduced, the data collection were not designed for the current study. Following, any results and conclusions must be clearly stated cautious, not generalizable and only hypothesis generating for further examination. Furthermore, the difference in social cognitive measures may not only contribute to inconsistent associations as stated - but also consistent associations may be a random effect, as we cannot be sure the same concepts are measured.

R1. We have further highlighted the partially exploratory and tentative nature of our current findings. We now spend greater time discussing these issues raised by the reviewer, in our discussion section, as quoted below.

“Given these limitations, our findings should be interpreted as preliminary. Future research should be conducted to confirm the present results in a large sample with data designed specifically to assess relations among psychopathology dimensions, GCA, and social cognitive ability (across multiple domains in the same participants). This would allow us to determine whether the present findings are robust and generalizable.”

2. I do miss an attempt to clarify in post hoc testing the more specific dynamic effect of GCA in the relation between p-factor and social cognition. Is it potentially mediating or moderating the effect?

R2. In our original findings, we did test for a statistical mediation effect and in fact found evidence for significant indirect effects of p-factor on social cognition through GCA (whereas direct effects of Detachment on social cognition were significant). Technically, we did not find statistical mediation (as typically discussed), as the total effect of p-factor on social cognition was not significant. We also wanted to avoid using strong language around “mediation”, given that many individuals interpret this language causally but our current study design is not causally informative. Refer to the “direct” vs. “indirect” effects in Tables 7 and 8 to see results relevant to statistical mediation. 

We did not originally test for interactions, given the large number of variables in our dataset and limitations of MPlus for simultaneously estimating exploratory measurement factors and interaction effects. We have added post-hoc, exploratory analyses to test for interactions, as noted below.

“[F]ollow-up analyses were conducted for each sample. First, we conducted post-hoc, exploratory moderation analyses to examine whether any psychopathology variables significantly interacted with GCA when statistically predicting social cognition. To do so, we extracted estimated factor scores for psychopathology dimensions and GCA (as well as for Social Cognition in the P-HCP dataset) using the regression method in Mplus. Interaction terms were then computed for each psychopathology dimension by multiplying corresponding factor-score estimates with the factor-score estimates for GCA. Then, we conducted regression analyses predicting social cognition using GCA, each psychopathology factor score, and each of the GCA-by-psychopathology interaction terms.”

“In Sample 1, there were no significant interaction effects of psychopathology and GCA, in statistically predicting social cognition. In Sample 2, there was a significant negative interaction between GCA and dependability in statistically predicting social cognition (β = -.108, p = .019); given that both GCA and dependability positively predicted social cognition, a negative interaction indicates an attenuation of these positive effects given increases in the other respective variable.”

3. The should be separate analyses for sex, as this may be important for both psychopathology and social cognition.

R3. We have added additional analyses controlling for sex, which are now incorporated into our methods, results, and discussion, as quoted below. 

“[F]ollow-up analyses were conducted for each sample… [W]e conducted post-hoc regression analyses using… extracted factor scores to confirm whether key findings remained significant when controlling for sex. In these analyses, sex was included as a dummy-coded predictor variable (females = 0, males = 1).”

“In both samples, all effects identified as significant in our primary ESEM analyses remained significant when controlling for sex (and directions of effects remained consistent). When accounting for GCA and psychopathology, Females showed better social cognition than males in both Samples 1 (β = -.067, p = .007) and 2 (β = -.107, p = .026).”

“In addition to these primary contributions, our findings also echo previous work that shows superior social cognitive ability in females… and offer preliminary findings to inform future research on how sex and psychopathology may interact to explain variance in social cognition.”

A few minor comments:

4. Rhetorical questions in an introduction is a really bad style.

R4. We have revised our first paragraph to now avoid use of rhetorical questions.

“Humans are inherently social, yet there is significant variation in our ability and motivation to form and maintain relationships. Some individuals navigate complex social situations—like networking, dating, and interviewing—with ease, while others find even simple interactions—such as visiting a coffee shop or making a phone call—a challenge. Investigating individual differences in the mechanisms that facilitate social interaction provides insight into this variation. For instance, social cognitive processes—including emotion recognition and mentalizing—enable individuals to recognize and interpret the mental and emotional states of others. These processes are crucial for adaptive functioning; however, individuals with mental illness or at higher risk of psychopathology often experience difficulties in one or more of these social cognitive domains.”

5. ASD is not a psychiatric disorder, but a neurodevelopmental disorder. The distinction is important for ASD subjects.

R5. We realize that ASD is not strictly considered a psychiatric disorder and have revised our language accordingly. 

“Research has consistently reported social cognitive deficits across a broad range of psychiatric and neurodevelopmental disorders—including schizophrenia, personality disorders, anxiety and depression, and autism spectrum disorder [1–5].”

Reviewer #2: PONE-D-24-24635

Constructs Across a Hierarchical, Dimensional Model of Psychopathology Show Differential Associations with Social and General Cognitive Ability

EVALUATION

This is really interesting research addressing the relationships between social cognition and psychopathology studying two samples (Psychosis Human Connectome Project, P-HCP, and Twin cities –community sample).

Importantly, the role of cognitive ability (aka, intelligence) is also considered.

Furthermore, a latent variable approach is applied here.

One of the key findings is that the relationship between social cognition and the p factor can be accounted for by cognitive ability, while detachment is related with social cognition even after controlling for cognitive ability.

Tables 7 and 8 depict important findings that deserve attention.

Comparison among values obtained from both samples are highly informative. Thus, for instance, the -.30 value between general cognitive ability and the p factor in the P-HCP samples is in high contrast with the null value for this relationship in the Twin cities sample.

The general picture provides important information.

The Ms. is well written and goes to the heart of the main research questions.

Therefore, I recommend publication.

R6. We are excited to hear Reviewer 2’s positive reception of our work! We hope that our revisions in response to Reviewer 1 have made the paper even stronger. 

In addition to these revisions in response to the reviewers, we have added the following statement regarding funding, as requested:

“The funders had no role in data collection or analysis, decision to publish, or preparation of the manuscript. In reference to Sample 1, the NIMH did play some role in setting forth the study aims and study design—though not for these particular research questions and analyses—given that data collection was funded through a U01 cooperative agreement. For Sample 2 and for our particular set of analyses, the funders had no role in study design, data collection, analysis, decision to publish, or preparation of the manuscript.”

---

## [Editor Report · Decision Letter 1]

27 Dec 2024

Constructs Across a Hierarchical, Dimensional Model of Psychopathology Show Differential Associations with Social and General Cognitive Ability

PONE-D-24-24635R1

Dear Dr. Blain,

We’re pleased to inform you that your manuscript has been judged scientifically suitable for publication and will be formally accepted for publication once it meets all outstanding technical requirements.

Kind regards,

Soo-Hee Choi

Academic Editor

PLOS ONE

---

## [Editor Report · Acceptance letter]

9 Jan 2025

PONE-D-24-24635R1 

PLOS ONE

Dear Dr. Blain, 

I'm pleased to inform you that your manuscript has been deemed suitable for publication in PLOS ONE. Congratulations! Your manuscript is now being handed over to our production team.

Kind regards, 

on behalf of

Dr. Soo-Hee Choi 

Academic Editor

PLOS ONE